# A Low-FODMAP Diet Provides Benefits for Functional Gastrointestinal Symptoms but Not for Improving Stool Consistency and Mucosal Inflammation in IBD: A Systematic Review and Meta-Analysis

**DOI:** 10.3390/nu14102072

**Published:** 2022-05-15

**Authors:** Ziheng Peng, Jun Yi, Xiaowei Liu

**Affiliations:** 1Department of Gastroenterology, Xiangya Hospital, Central South University, Changsha 410008, China; pengzh96@csu.edu.cn; 2Hunan International Scientific and Technological Cooperation Base of Artificial Intelligence Computer-Aided Diagnosis and Treatment for Digestive Disease, Changsha 410008, China; 3National Clinical Research Center for Geriatric Disorders, Xiangya Hospital, Central South University, Changsha 410008, China

**Keywords:** low-FODMAP diet, inflammatory bowel disease, functional gastrointestinal symptoms, meta-analysis, mucosal inflammation

## Abstract

Background: A low fermentable oligosaccharides, disaccharides, monosaccharides, and polyols diet (LFD) is claimed to improve functional gastrointestinal symptoms (FGSs). However, the role of LFD in inflammatory bowel disease (IBD) patients with FGSs remains unclear. Objective: To systematically assess the efficacy of LFD in IBD patients with FGSs. Methods: Six databases were searched from inception to 1 January 2022. Data were synthesized as the relative risk of symptoms improvement and normal stool consistency, mean difference of Bristol Stool Form Scale (BSFS), Short IBD Questionnaire (SIBDQ), IBS Quality of Life (IBS-QoL), Harvey-Bradshaw index (HBi), Mayo score, and fecal calprotectin (FC). Risk of bias was assessed based on study types. A funnel plot and Egger’s test were used to analyze publication bias. Results: This review screened and included nine eligible studies, including four randomized controlled trials (RCTs) and five before–after studies, involving a total of 446 participants (351 patients with LFD vs. 95 controls). LFD alleviated overall FGSs (RR: 0.47, 95% CI: 0.33–0.66, *p* = 0.0000) and obtained higher SIBDQ scores (MD = 11.24, 95% CI 6.61 to 15.87, *p* = 0.0000) and lower HBi score of Crohn’s disease (MD = −1.09, 95% CI −1.77 to −0.42, *p* = 0.002). However, there were no statistically significant differences in normal stool consistency, BSFS, IBS-QoL, Mayo score of ulcerative colitis, and FC. No publication bias was found. Conclusions: LFD provides a benefit in FGSs and QoL but not for improving stool consistency and mucosal inflammation in IBD patients. Further well-designed RCTs are needed to develop the optimal LFD strategy for IBD.

## 1. Introduction

Inflammatory bowel disease (IBD), including ulcerative colitis (UC) and Crohn’s disease (CD), is a group of chronic gastrointestinal diseases with frequent functional gastrointestinal symptoms (FGSs), such as abdominal pain and bloating. Prior preliminary cross-sectional research works reveal an overall prevalence of FGSs in 30% to 45% of IBD outpatients and a negative impact on both psychological wellbeing and quality of life (QoL) in the presence of these symptoms [1]. Higher rates of anxiety and depression and lower QoL scores were reported consistently in IBD patients with FGSs [1,2,3]. The etiology of these FGSs in IBD remains unclear, and the gastrointestinal damage or the psychological impact of IBD may be partially responsible in the process [4]. 

Over recent decades, dietary structure has increasingly garnered the attention of the public for its prominent roles in IBD. The dietary structure itself is assumed to cause intestinal inflammation. A variety of FGSs can be triggered by the form and nutrient content of ingested food through a matrix of different mechanisms, including bacterial fermentation altering gut microbiota, the induction of distinct osmotic load effects in the small bowel and colon, the production of gas in the gastrointestinal tract, and the activation or suppression of immune responses [5,6]. It has been proposed that certain dietary habits, such as the intake of food rich in dietary fiber or lactose free, may influence the FGSs. However, these benefits of diet have not yet been confirmed [7,8]. Thus, it remains questionable whether adjustment of dietary structure and dietary intervention has beneficial effects on FGSs of IBD. Recently, great concern has been placed on the dietary restriction of fermentable oligosaccharides, disaccharides, monosaccharides, and polyols (FODMAP) due to their poor absorption and unique gradual roles in the gastrointestinal tract. In irritable bowel syndrome (IBS), a low-FODMAP diet (LFD) has been shown to ameliorate FGSs by reducing diet-induced luminal water, colonic gas, and, consequently, luminal-distension-induced visceral hypersensitivity [9,10]. Hence, LFD arouses an increasing interest internationally and has been proposed as one of the symptomatic therapies for IBS [11] and a complementary regime alleviating symptoms of functional gastrointestinal disorders [12].

More interestingly, IBD patients with FGSs, even in remission, were found to be more frequently prone to dysbiosis [13], chronic relapse phases [4], a compromised immune response, increased gut permeability, and a discorded brain–gut axis than those without FGSs [14]. The dietary intervention has been claimed to provide symptom improvement during the acute and chronic stages of IBD [15]. It can mitigate disease progression or avert potentially disastrous complications by altering the microbiota, metabolome, host barrier function, and innate immunity [16,17]. Studies have revealed that a semi-vegetarian diet has a preventive effect of diminishing symptoms, delaying disease progression, improving QoL, and preventing relapse of CD [18]. In a cross-over study of patients with UC in remission, a low-fat, high-fiber diet and an improved standard American diet (including higher quantities of fruits, vegetables, and fibers than a typical standard American diet) were found to increase QoL; in addition, the low-fat and high-fiber diet decreased markers of inflammation and reduced intestinal dysbiosis in fecal samples [19]. Likewise, as a dietary intervention program, LFD is a widespread concern and has been explored in IBD. It is reasonable to speculate that LFD could provide benefits for FGSs, QoL, and intestinal inflammation in IBD. Several studies assessing the efficacy of LFD in IBD have been conducted worldwide [20,21,22,23,24,25,26,27,28], yet with controversial results being reported.

Five years ago, a meta-analysis had been performed to identify the role of LFD in IBD [29]. However, the study focused only on FGSs and did not investigate the efficacy of LFD on QoL, disease activity, and inflammatory markers in IBD patients. We performed an updated and more comprehensive meta-analysis and aimed to reveal the efficacy of LFD on the Gastrointestinal Symptom Rating Scale (GSRS), the Irritable Bowel Syndrome–Symptom Severity System (IBS-SSS), stool consistency, the Short IBD Questionnaire (SIBDQ), the IBS Quality of Life (IBS-QoL) questionnaire, Harvey-Bradshaw index (HBi) for CD, Mayo score for UC, and fecal calprotectin (FC) in IBD patients.

## 2. Methods

This systematic review and meta-analysis were conducted under the guidance of the Preferred Reporting Items for Systematic Reviews and Meta-Analysis (PRISMA) protocols [30]. In addition, the protocol was registered in the International Prospective Register of Systematic Reviews (PROSPERO) (registration number CRD42022302018).

### 2.1. Literature and Search Strategy 

Two investigators (ZP and JY) independently conducted a systematic literature retrieval in PubMed, Web of Science, EMBASE, Cochrane Central Register of Controlled Trials, Chinese National Knowledge Infrastructure (CNKI), WanFang (Chinese) Database, with the last search update on 1 January 2022. Studies were screened without geographical and language restrictions. The search terms for retrieval in these databases were: ‘FODMAP’ OR ‘FODMAPS’ OR ‘Fermentable, poorly absorbed, short-chain carbohydrates’, OR ‘Fermentable oligosaccharides, disaccharides, monosaccharides and polyols’, AND ‘inflammatory bowel disease’ OR ‘IBD’ OR ‘Crohn’s disease’ OR ‘CD’ OR ‘ulcerative colitis’ OR ‘UC’ or equivalent terms. Additionally, the retrieved references were screened manually to find the relevant potential literature.

### 2.2. Literature Screening

Eligible studies fulfilling the following criteria were included in our meta-analysis: (1) all relevant randomized controlled trials (RCTs) or before–after studies in the same patient; (2) a definitively established IBD diagnosis; (3) comparing LFD with a placebo diet or a usual diet (hereafter referred to as normal diet (ND)) or comparing pre- and post-contrast LFD; and (4) outcomes including overall and individual FGSs response, SIBDQ, IBS-QoL, GSRS, stool consistency, Mayo score for UC, HBi for CD, and FC. The exclusion criteria were presented as follows: (1) participants suffering from other digestive disorders; and (2) participants receiving multiple interventions simultaneously. Literature screening was carried out by two independent investigators (ZP and JY), and a third investigator (XL) resolved disagreements.

### 2.3. Data Extraction

From each included study, the following information was collected by two independent investigators (ZP and JY) using a standardized data extraction form: (1) general information: title, the first author, publication year, and the country of the study; (2) study information: study design, participants, intervention, duration of therapy, and outcome evaluation of FGSs; (3) baseline characteristics: total case/controls or cohort size, age range or mean age (standard deviation, SD), sex, and type of IBD; (4) outcomes: the number or percentage of patients with overall and individual FGSs improvement and normal stool consistency before and after the intervention; the mean difference (MD) of GSRS, IBS-SSS, SIBDQ, IBS-QoL, Bristol Stool Form Scale (BSFS), Mayo score for UC, HBi for CD, and level of FC. One investigator (ZP) was responsible for contacting the original author for complete data. All investigators participated in the discussion to resolve the dispute. 

### 2.4. Assessment of Risk of Bias

The risk of bias assessment in the included RCTs was performed by two independent investigators using the Cochrane Risk of Bias Tool and Jaded scale with Review Manager (RevMan) (Version 5.3, Cochrane Collaboration, Oxford, UK) [31]. The investigators evaluated the quality of the non-RCTs included, according to the Methodological Index for Non-Randomized Studies (MINORS) [32]. The best possible score on this scale is 16 points. 

### 2.5. Statistical Analysis

Data analysis was performed using RevMan 5 (Version 5.3, Cochrane Collaboration, Oxford, UK). Firstly, a chi-square test and the *I*^2^ statistic were used to assess the heterogeneity of each study included. Among them, *I*^2^ statistic means the percentage of total variability due to heterogeneity between studies. Secondly, according to the result of heterogeneity, the appropriate Mantel–Haenszel fixed-effects model or the DerSimonian and Laird random-effects model were selected to calculate the MD with 95% confidence intervals (CIs) for continuous data and the risk ratio (RR) with 95% CIs for dichotomous data. The random-effects model was used in case of high heterogeneity (*p* < 0.10 or *I*^2^ statistic value > 50%); otherwise, the fixed-effects model was used (*p* > 0.10 or *I*^2^ statistic value < 50%). In addition, a funnel plot and the Egger’s test (using the Stata 15 software) were used to assess publication bias with a *p* value of <0.10 indicating statistical significance. 

## 3. Results

### 3.1. Characteristics of Eligible Studies

Initially, 476 articles were identified and screened via reviewing the title, abstract, and full text. Then, 467 articles were excluded for various reasons, such as non-human or non-original research or article with incomplete information, etc. Finally, nine studies were included for estimating the effect of LFD on IBD patients. A flowchart shown in Figure 1 presents the details of included studies and the selection process.

Eventually, our meta-analysis was performed based on the inclusion of three prospective studies [20,21,22], one retrospective study [23], one study that included both prospective and retrospective components [24], and four RCTs [25,26,27,28], with a total of 446 IBD patients. Specifically, the five non-RCTs involved 256 IBD patients, and the four RCTs randomly divided IBD patients into the experimental group and the control group, involving 190 patients. Among them, IBD patients with ND, used as the control, contained 95 cases (nine patients participating in the cross-over trials were included in LDF group). Therefore, nine studies involving 351 LFD cases and 95 controls were analyzed in this meta-analysis. The baseline characteristics and data extraction from the included studies are outlined in Table 1 and Table 2.

### 3.2. Overall Symptom Response 

Considering the inconsistent definition standards for FGS improvement in different studies generally (Table 2), we analyzed the number of people suffering from FGSs before and after LFD intervention in non-RCTs or the LFD group and ND group in RCTs. As a whole, all the nine studies showed that LFD was associated with an improvement of FGSs in IBD patients (RR: 0.47, 95% CI: 0.33–0.66, *p* = 0.0000) (Figure 2a).

No difference was found in the subgroups classified by disease type. Symptom improvement was significant in both CD patients (RR: 0.44, 95% CI: 0.34–0.55, *p* = 0.0000) and UC patients (RR: 0.43, 95% CI: 0.33–0.56, *p* = 0.0000) (Figure 2b).

### 3.3. Individual Symptom Response

In terms of individual FGSs, the greatest improvement occurred in bloating (RR: 0.37, 95% CI: 0.24–0.57, *p* = 0.0000), followed by wind or flatulence (RR: 0.38, 95% CI: 0.28–0.51, *p* = 0.0000), borborygmi (RR: 0.48, 95% CI: 0.26–0.89, *p* = 0.02), abdominal pain (RR: 0.5, 95% CI: 0.37–0.68, *p* = 0.0000), and fatigue or lethargy (RR: 0.71, 95% CI: 0.61–0.82, *p* = 0.000). However, no significant difference was found in nausea or vomiting (RR: 0.54, 95% CI: 0.22–1.32, *p* = 0.18) between the LFD group and ND group (Figure 3).

### 3.4. Degrees of Change in FGSs

Three studies assessed FGS changes using GSRS as the continuous variable, showing that LFD was associated with a reduction in total GSRS score (MD = −0.43, 95% CI −0.54 to −0.33, *p* = 0.000) (Figure 4a). Meanwhile, two studies assessed FGS changes using IBS-SSS, showing that LFD was associated with a reduction in total IBS-SSS score (MD = −93.37, 95% CI −144.33 to −42.42, *p* = 0.003) (Figure 4b).

### 3.5. QoL Score

Only two studies assessed the QoL score using SIBDQ, showing that LFD was associated with a reduction in total SIBDQ score (MD = 11.24, 95% CI 6.61 to 15.87, *p* = 0.0000) (Figure 4c). Interestingly, two studies evaluated QoL using the IBS-QoL without observing a significant difference (MD = −3.73, 95% CI −22.19 to 14.74, *p* = 0.69) (Figure 4d).

### 3.6. Stool Consistency

The Bristol Stool Chart (BSC) illustrates seven different stool types representing constipation (type 1–2), normal stools (type 3–4), and diarrhea (type 5–7) [33,34]. Two studies reported normal stool consistency (type 3–4) as dichotomous outcomes, with no significant difference observed (RR: 5.99, 95% CI: 0.17–216.51, *p* = 0.33) between the LFD group and ND group (Figure 5a). Similarly, two studies assessed stool consistency using BSFS, showing no significant difference between the LFD group and ND group (MD = −0.17, 95% CI −0.48 to 0.15, *p* = 0.30) (Figure 5b).

### 3.7. Disease Activity

For UC, two studies reported the Mayo score, yet with no difference between the LFD group and ND group (MD = −0.32, 95% CI −1.09 to 0.45, *p* = 0.41) (Figure 5c). In contrast, three studies showed that LFD was associated with a reduction in HBi score for CD (MD = −1.09, 95% CI −1.77 to −0.42, *p* = 0.002) (Figure 5d).

### 3.8. FC

FC was analyzed using the synthesis from three studies, showing no significant changes (MD = −16.03, 95% CI −36.78 to 4.73, *p* = 0.13) (Figure 5e).

### 3.9. Quality of the Included Studies

The overall risk of bias of four included RCTs was relatively low, as shown in Figure 6. Meanwhile, for the remaining five non-RCTs, according to the MINORS, four studies scored 14 points, and one study scored 12 points (Table 1).

### 3.10. Publication Bias

No evidence of publication bias was found based on Egger’s regression test, i.e., overall symptom response (*p* = 0.997), overall symptom response of CD (*p* = 0.871), overall symptom response of UC (*p* = 0.895), abdominal pain (*p* = 0.827), nausea/vomiting (*p* = 0.106), bloating (*p* = 0.771), wind/flatulence (*p* = 0.464), borborygmi (*p* = 0.549), fatigue/lethargy (*p* = 0.273), GSRS (*p* = 0.633), HBi for CD (*p* = 0.652), and FC (*p* = 0.799). In addition, the shape of the funnel plot also suggested no evident publication bias (Figure 7).

## 4. Discussion

This updated systematic review and meta-analysis included four RCTs and five before–after studies, with 446 participants in total. The study aimed to pool data from existing studies to examine whether LFD alleviates FGSs effectively in IBD patients. Additional data were extracted from existing studies and were used to uncover the efficacy of LFD on SIBDQ, IBS-QoL, stool consistency, Mayo for UC, HBi for CD, and FC in IBD patients. The present study found that LFD alleviated FGS, obtained higher SIBDQ scores, and reached remission or low disease activity in CD. However, there were no statistically significant differences in the efficacy of LFD on the IBS-QoL, stool consistency, Mayo for UC, and FC in IBD patients. It is worth noting that, in addition to assessing FGSs, the number of original studies and participants included is small. The credibility of these results remains unexamined, which requires large-scale clinical trials for further confirmation. Collectively, this meta-analysis suggested that IBD patients with FGSs may profit from LFD treatment with the assistance of a healthcare professional.

The primary outcome of this study was that LFD can improve FGSs in IBD. Symptom improvement was observed in bloating, wind or flatulence, borborygmi, abdominal pain, and fatigue or lethargy in IBD patients, except for nausea and vomiting. No difference in symptom improvement was found in patients with different subtypes, since LFD resulted in similar results of FGS alleviation in both CD and UC patients. The evidence that both IBS-SSS and GSRS scores significantly decreased in patients with LFD intervention further supports the effect of LFD. Water in the small intestinal increases through osmotic potential by absorbing fermentable carbohydrates, such as fructose and mannitol. Intestinal gas (wind) increases through fermenting food by intestinal bacteria, such as fructans and galacto-oligosaccharides [27]. Increased intestinal water and gas appear to play an integral role in triggering symptoms of IBS, such as bloating, abdominal pain, excessive flatus, and altered bowel habit [35]. Approximately 20–60% of IBS patients complained that some food elements trigger their FGSs, especially the ‘gas-producing’ food (e.g., dairy products, certain fruits, wheat, pulses and legumes, cruciferous vegetables, etc.), with symptoms improved when removing these food items from their diet. Indeed, food hypersensitivity, food allergy, food intolerance, and nonceliac gluten sensitivity are considered to be responsible for these food-related symptoms [36]. In consideration of the proposed mechanism of action of LFD, the top three greatest beneficial symptoms were bloating, flatulence, and borborygmi, which was consistent with previous studies and our meta-analysis in both IBS and IBD [25,35,37,38]. It is noteworthy that no difference was found in nausea or vomiting between groups. The result may be explained by continuous immune activation after LDF intervention, since, compared to other FGSs, nausea or vomiting is the symptom most indicative of the elevated level of interleukin-2 [39].

IBD is a chronic relapsing–remitting gastrointestinal disease. Treatment for IBD consists of diminution or elimination of disease activity and optimization of health-related QoL [40]. The uncertainty of the symptoms and the unpredictability of this clinical condition is highly demanding for IBD patients and deteriorates their QoL [41,42]. Thus, it is obvious that the QoL of patients may be affected by the disease course (extent, severity, and pattern of symptoms’ relapse), prescribed therapy (efficacy, side effects, and burden of administration), and psychosocial factors [40,43,44,45]. IBD patients who suffer from FGSs are more likely to experience anxiety and depression [46]. In the present study, there was a conflicting conclusion of the efficacy of LFD on QoL: a decreased SIBDQ score and an IBS-QoL score with no significant difference. One reason for this may be that IBS-QoL focuses on the impact of stool output, while SIBDQ centers on the multifactorial impact, such as psycho-emotional functioning, systemic symptoms, bowel symptoms, and social functioning. In agreement with this expectation, no beneficial effects were observed in the stool consistency. Thus, it is undeniable that LDF positively affects FGS, and the QoL may be influenced by FGSs in IBD patients. 

Stool consistency commonly refers to the rheology or viscosity of the stool, which is strongly dependent on the stool water content [47]. The BSFS is the most widely used scale to quantify stool consistency [48,49]. Diarrhea is the hallmark and the first symptom associated with IBD and appears in 77% of UC patients and 82% of CD patients [50]. The pathogenesis of IBD-associated diarrhea is essentially an outcome of mucosal damage caused by persistent inflammation. Altered expression and/or function of epithelial ion transporters and channels cause electrolyte retention and water accumulation in the intestinal lumen of IBD patients. In addition, aberrant barrier function further contributes to diarrhea via the leak–flux mechanism [51]. Our results do not suggest significant improvement in stool consistency, in terms of both the BSFS score and the number of normal stool consistency (type 3–4 of BSC) after LFD, suggesting no improvement in persistent mucosal inflammation of IBD by LFD.

By comparison, in the aspect of clinical remission, more importance needs to be attached to mucosal healing in IBD management, for the latter predicted a durable complete remission [52]. Several studies have found that both the Crohn’s Disease Activity Index (CDAI) and HBi had low specificity and did not correlate well with the endoscopic or histological disease activity of CD patients [53,54]. A considerable proportion of patients who reported clinical remission have mucosal inflammation [55]. The Mayo score is deemed to be more reliable for assessing disease activity of UC patients, for it includes endoscopic score and physicians’ clinical assessments, while the CDAI or HBi score attaches more attention to subjective symptoms of CD patients [56]. It is worthy of note that FC is now widely recommended as a sign of intestinal mucosal healing [57]. Low FC has been demonstrated to predict sustained clinical remission in IBD patients [58]. According to our study, pooled data showed a decrease in HBi score and no significant difference in the Mayo score and level of FC. Nevertheless, it should be noted that the study subjects remained in clinical remission in a majority of the included articles. Whether a slight reduction in HBi score in our study can truly reflect a variation in disease activity is yet to be determined. Additionally, it remains to be shown if there is a ceiling effect created by low disease activity on improvement of inflammation by LFD. Hence, further research with a larger sample size and more comprehensive analysis is warranted to validate our results.

The limitations of our study are as follows. Firstly, there were different evaluation standards to assess the relief of FGSs among distinct researchers. With no unified standard, there could be controversial results in different studies, while in the included RCTs, the diets were not standardized and specified in the control group, which might have produced result bias due to the different dietary habits in different regions. Secondly, there was a relatively smaller sample size of the non-RCTs and RCTs included in this meta-analysis. This potentially reduces the reliability of the results. The drawback is more pronounced in subgroup analyses. Additional studies on this topic should be developed to address this question. Thirdly, significant heterogeneity was found among the included studies, which may potentially impact the results of the meta-analysis. The cause of the heterogeneity is still unclear, which may be attributed to the inconsistent research methods and a relatively small number of primary studies. To deal with the potential heterogeneity and provide quality evidence, we performed subgroup analysis according to the statistical method and used a random-effect model suggested by Liberati et al. [59]. Despite the above, it is still the most comprehensive and rigorous meta-analysis to date. Detailed subgroup analysis would benefit basic researchers the most, and no publication bias was found during analysis.

It should be emphasized that despite these improvements in FGSs, intervention using the LFD in IBD should be carefully considered and closely monitored. Indeed, the included studies did not report general or severe adverse effects of the short-term LFD intervention. However, it is important to take into consideration patient adherence and the risk of compromising nutritional status with a long-term restrictive diet. It is well known that undernutrition is common in IBD. Therefore, the use of restrictive diets should be supervised by a dietitian [60], associated with the monitoring of vitamin and mineral deficiencies and proper supplementation accordingly [61]. 

## 5. Conclusions

In conclusion, our meta-analysis demonstrates that LFD has a favorable role in alleviating FGSs in IBD patients, yet without significant benefits in improving stool consistency and mucosal inflammation. Consuming LFD based on professional advice from health care professionals is recommended for IBD patients with problematic FGSs, especially those in remission. Moreover, well-designed and large-scale RCTs are required in the future to confirm the findings and develop the optimal LFD strategy for IBD.

## Figures and Tables

**Figure 1 nutrients-14-02072-f001:**
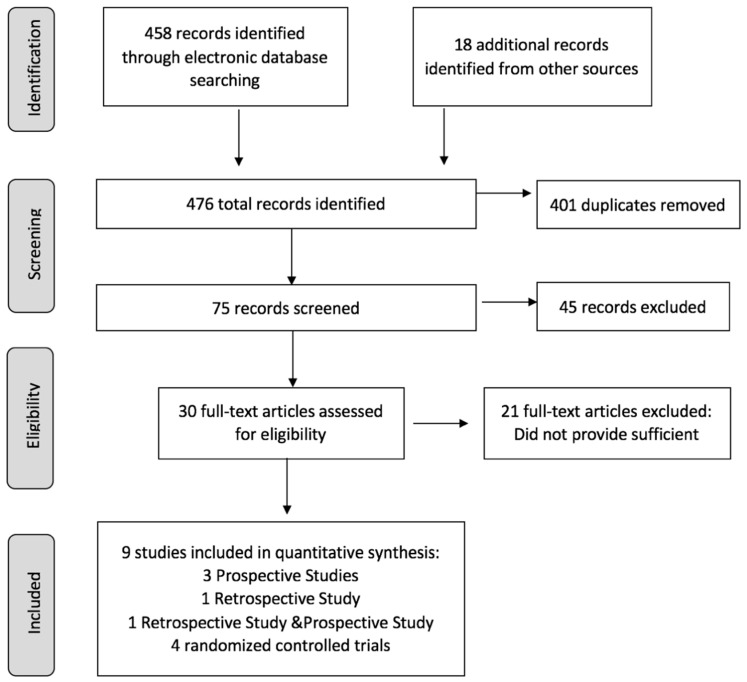
Flowchart summarizing the research and selection of articles for the meta-analysis.

**Figure 2 nutrients-14-02072-f002:**
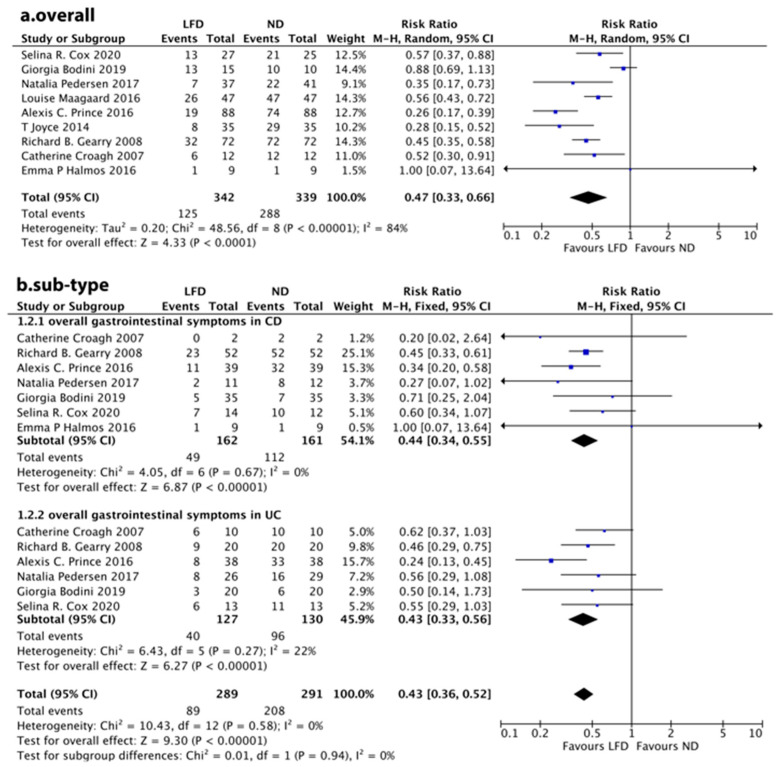
Meta-analysis for a low-FODMAP diet and (**a**) overall gastrointestinal symptoms in patients with inflammatory bowel disease, (**b**) gastrointestinal symptoms in patients with Crohn’s disease or ulcerative colitis.

**Figure 3 nutrients-14-02072-f003:**
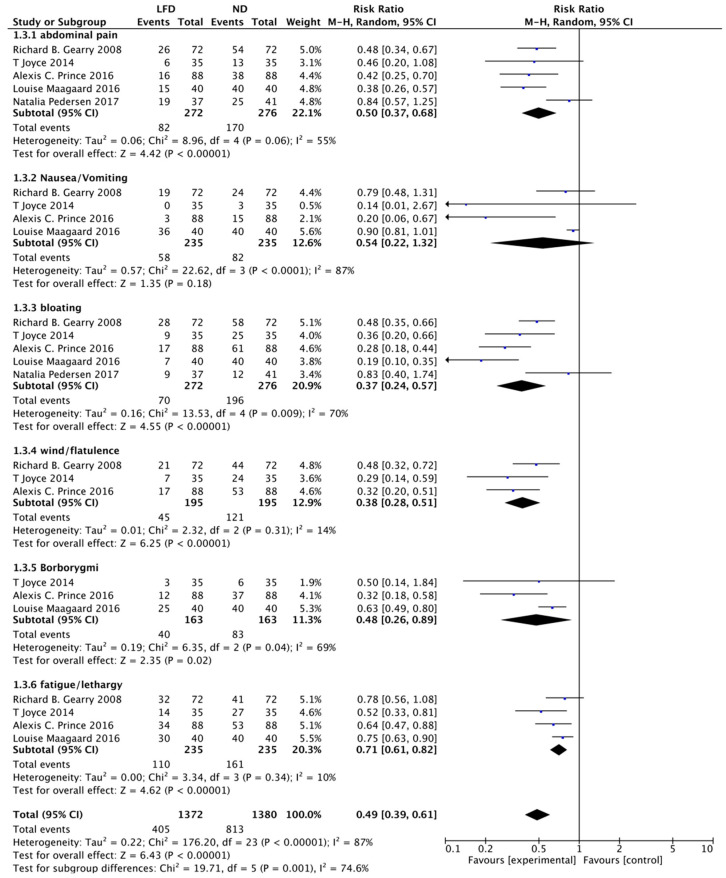
Meta-analysis for a low-FODMAP diet and individual gastrointestinal symptoms in patients with inflammatory bowel disease.

**Figure 4 nutrients-14-02072-f004:**
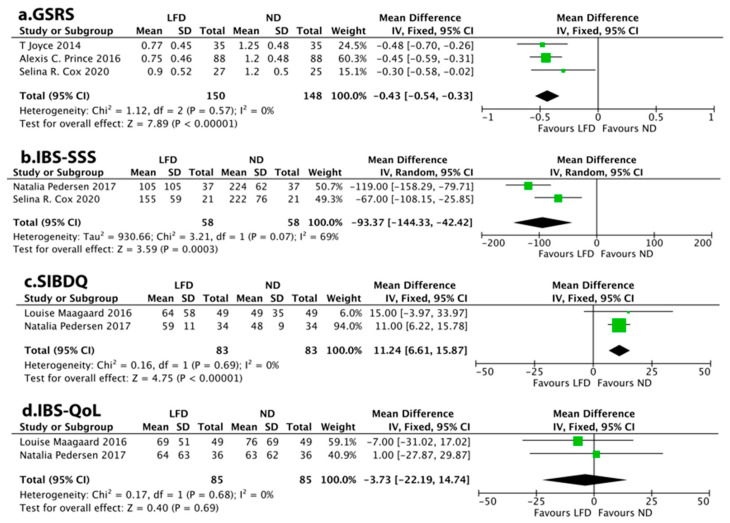
Meta-analysis for a low-FODMAP diet and (**a**) the Gastrointestinal Symptom Rating Scale (GSRS), (**b**) the Irritable Bowel Syndrome–Symptom Severity System (IBS-SSS), (**c**) the Short IBD Questionnaire (SIBDQ), (**d**) the IBS Quality of Life (IBS-QoL) questionnaire in patients with inflammatory bowel disease.

**Figure 5 nutrients-14-02072-f005:**
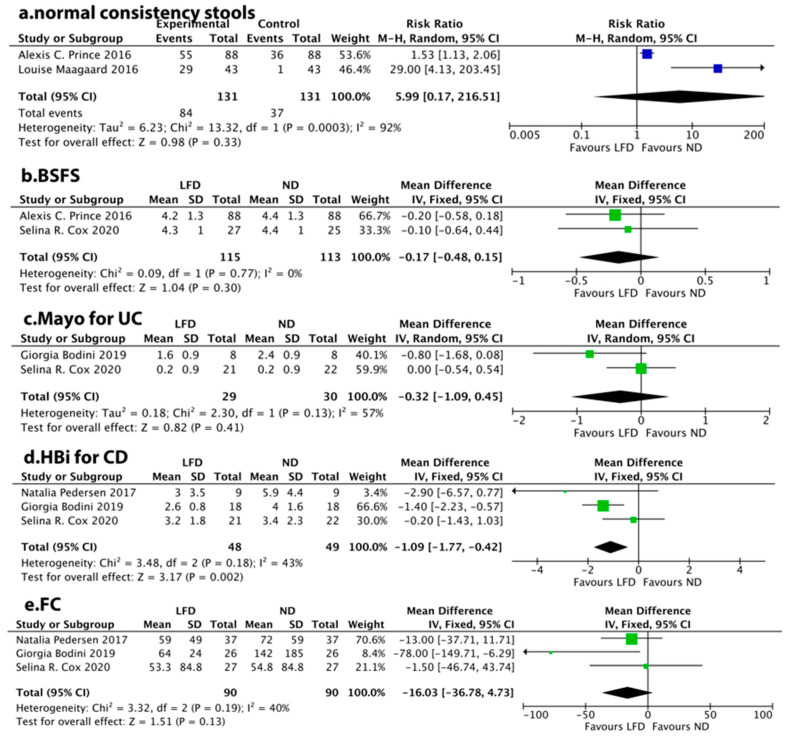
Meta-analysis for a low-FODMAP diet and (**a**) normal stool consistency, (**b**) the Bristol Stool Form Scale (BSFS), (**c**) Mayo score in patients with ulcerative colitis, (**d**) Harvey-Bradshaw index (HBi) score in patients with Crohn’s disease, (**e**) fecal calprotectin (FC).

**Figure 6 nutrients-14-02072-f006:**
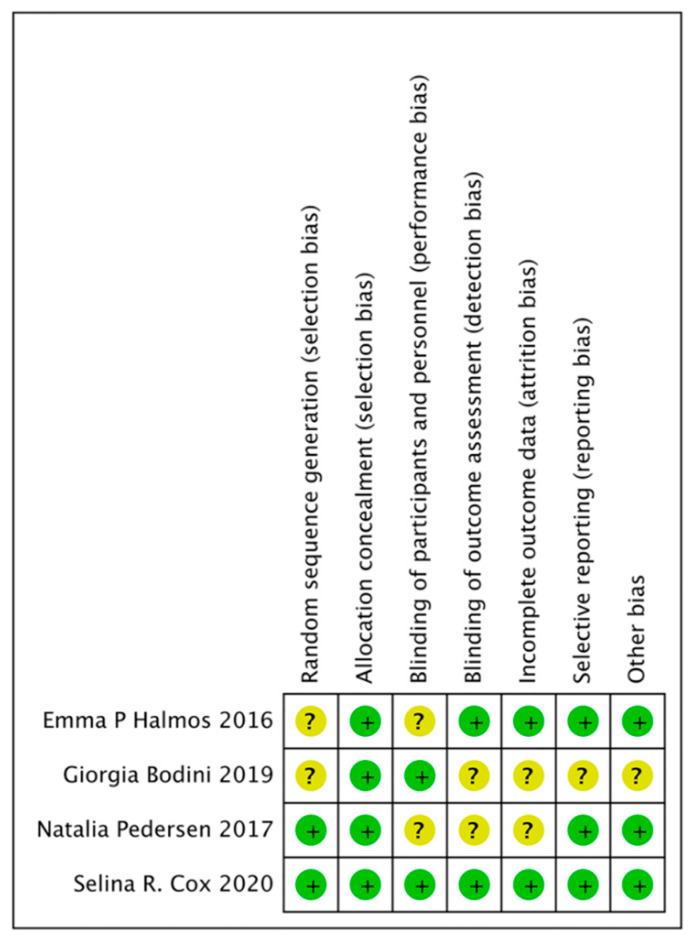
Risk of bias summary of the four included randomized controlled trials (RCTs) [25,26,27,28]. (+, high-risk; ?, uncertain).

**Figure 7 nutrients-14-02072-f007:**
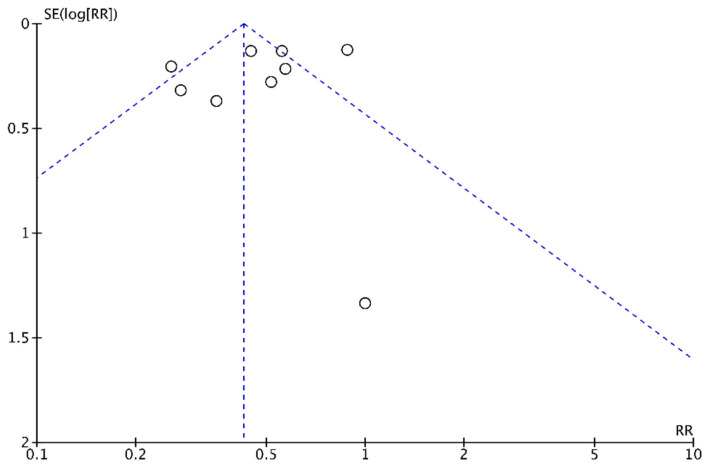
Funnel plot of the low-FODMAP diet and gastrointestinal symptoms. No evidence of publication bias was found based on Egger’s regression test (*p* = 0.997).

**Table 1 nutrients-14-02072-t001:** Baseline characteristics of the included studies in this meta-analysis.

Study	Design	Total Case (Controls or Cohort Size)	Age Range or Mean Age (SD)/Year	Male/Female	Type of IBD (CD/UC/IBD-u)	MINORS Scores
Alexis C. Prince et al.	Non-RCT	88	20–80	26/62	39/38/11	14
Richard B. Gearry et al.	Non-RCT	72	18–72	39/33	52/20/0	14
T. Joyce et al.	Non-RCT	35	39(NA)	13/22	17/17/1	12
Louise Maagaar et al.	Non-RCT	49	19–70	9/40	32/12/5	14
Catherine Croagh et al.	Non-RCT	12	35–74	5/7	2/10/0	14
Natalia Pedersen et al.	RCT	78(LFD:37 ND:41)	LFD:20–70 ND:24–69	LFD:12/32 ND:10/35 *	LFD:14/30/0 ND:14/31/0 *	-
Giorgia Bodini et al.	RCT	51(LFD:26 ND:29)	LFD:34–48 ND:44–57	LFD:7/19 ND:12/17	LFD:18/8/0 ND:17/12/0	-
Selina R. Cox et al.	RCT	52(LFD:27 ND:25)	LFD:33(11) ND:40(13)	LFD:10/17 ND:13/12	LFD:14/13/0 ND:12/13/0 *	-
Emma P. Halmos et al. ^#^	RCT	9(LFD:9 ND:9)	29–41	3/6	9/0/0	-

RCT: randomized controlled trial; IBD: inflammatory bowel disease; CD: Crohn’s disease; UC: ulcerative colitis; IBD-u: unknown type of inflammatory bowel disease; NA: not available; LFD: low-FODMAP diet; ND: normal diet. *: baseline characteristics without drops. ^#^: a randomized, controlled cross-over trial, and the experimental and control groups consisted of the same 9 subjects.

**Table 2 nutrients-14-02072-t002:** Data abstraction from the included studies.

Author	Year	Country	Participants	Intervention	Duration of Therapy	Outcome Evaluated FGS
Prospective Study						
Alexis C. Prince et al.	2016	United Kingdom	IBD patients with persistent FGS	Low FODMAP	6 Weeks	Primary outcome was assessment of satisfactory relief of FGS measured using GSQ. Individual symptoms were assessed using the GSRS.
Richard B. Gearry et al.	2008	Australia	IBD patients with persistent abdominal symptoms	Low FODMAP	3 Months	An arbitrary improvement of 5 or more on a custom gastrointestinal symptoms scale was used as a measure of unequivocal improvement for each symptom.
T. Joyce et al.	2014	United Kingdom	Patients with inactive IBD and FBD	Low FODMAP	6 Weeks	Symptoms were measured using the GSQ and the GSRS.
Retrospective Study						
Louise Maagaard et al.	2016	Denmark	Consecutive patients with IBD	Low FODMAP	6–8 Weeks	Patient-reported effectiveness of the low-FODMAP diet. Effectiveness was categorized as full, partial, or no effect.
Retrospective Study and Prospective Study						
Catherine Croagh et al.	2007	Australia	IBD with colectomy and ileal pouch formation or ileorectal anastomosis	Low FODMAP	6 Weeks	Patient-reported effectiveness of diet on symptoms. Effectiveness was categorized as improved, no change, or worse.
Randomized Controlled Trial						
Natalia Pedersen et al.	2017	Denmark	IBD patients with a baseline IBS-SSS of at least 75 points	Low-FODMAP or normal habitual diet	6 Weeks	Primary outcome was the number of patients achieving a 50-point reduction in IBS-SSS.
Giorgia Bodini et al.	2019	Italy	IBD patients in the remission phase or mild disease activity	Low-FODMAP or standard diet	6 Weeks	Patients with a total IBD-Q score >170 were assessed as being in symptomatic remission.
Selina R. Cox et al.	2020	United Kingdom	Adult quiescent IBD patients with ongoing gut symptoms	Low-FODMAP or placebo sham diet	4 Weeks	The global symptom question was used to assess adequate relief of FGS at end of trial.
Emma P. Halmos et al.	2016	Australia	Quiescent CD patients with stable therapy	Low or typical (Australian) FODMAP diets	21 Days	The visual analog scale score was used to measure overall gastrointestinal symptoms.

FGS: functional gastrointestinal symptoms; IBD: inflammatory bowel disease; GSRS: Gastrointestinal Symptom Rating Scale; FBD: functional bowel disorders; GSQ: the global symptom question (Do you currently have satisfactory relief of your gut symptoms?); IBS-SSS: Irritable Bowel Syndrome–Symptom Severity System; IBD-Q: inflammatory bowel disease—quality of life; CD: Crohn’s disease.

## Data Availability

Data available on request.

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
