# Peer review of "A Low-FODMAP Diet Provides Benefits for Functional Gastrointestinal Symptoms but Not for Improving Stool Consistency and Mucosal Inflammation in IBD: A Systematic Review and Meta-Analysis"

_nutrients, 2022, doi:10.3390/nu14102072_

Round 1

Reviewer 1 Report

The paper is really well written.

Some comments were raised to shape up the manuscript.

Please provide us a reply point by point on the comments addressed. 

Author Response

Dear Editor and Reviewers,

On behalf of my co-authors, we thank you very much for giving us an opportunity to revise our manuscript, we appreciate the editor and reviewers very much for their positive and constructive comments and suggestions on our manuscript entitled “A Low-FODMAP Diet Provides Benefits for Functional Gastrointestinal Symptoms But Not for Improving Stool Consistency and Mucosal Inflammation in IBD: A Systematic Review and Meta-Analysis”, with manuscript ID “nutrients-1711952”. The comments have been carefully taken into account and a newly revised submission has been uploaded. We highlighted all the altered passages in light yellow. The responses are as follows.

Response to the Reviewer 1,

Comment 1:

Introduction: Authors should add a hypothesis.

Reply 1:

Thank you for your comments so much. We are very sorry for our negligence of a hypothesis in Introduction. We have added the hypothesis in Introduction for easier understanding.

Changes in the text:

—See lines 77-79 in “Revised manuscript”.

Comment 2:

Authors should reply to this comment:Do the authors (inappropriately) discuss their own opinions and bias? Do the authors address gaps in clinical knowledge?

Reply 2:

Thank you for your critical comments. After discussion, all authors agree that there are some comments needed to be rephrased in the discussion on our own results. As you consider, a meta-analysis, as a binary study, is more important to present the results objectively. We have revised some sentences to present conflicting results and possible biases in this paper to guide readers to open-ended thinking and inspire a more rigorous research design.

Changes in the text:

—See lines 332-333 in “Revised manuscript”.

We have tried our best to revise our manuscript according to the comments. Attached please find the revised version, which we would like to submit for your kind consideration. 

We would like to express our great appreciation to you for your comments on our paper. Looking forward to hearing from you. Thank you and best regards.

Yours sincerely,

Jun Yi, MD, PhD & Xiaowei Liu, MD, PhD

Institute:

  1. Department of Gastroenterology, Xiangya Hospital of Central South University, Changsha, Hunan, 410008, China.
  2. Hunan International Scientific and Technological Cooperation Base of Artificial Intelligence Computer Aided Diagnosis and Treatment for Digestive Disease, Changsha, Hunan, 410008, China.
  3. National Clinical Research Center for Geriatric Disorders, Xiangya Hospital, Central South University, Changsha, Hunan, 410008, China.

Address:

Department of Gastroenterology, Xiangya Hospital of Central South University, Changsha, Hunan, 410008, China.

Tel:+86-731-89753268

Fax: +86-731-89753268

E-mail:junyee1989@csu.edu.cn & liuxw@csu.edu.cn

Reviewer 2 Report

This is an excellent review and meta analysis assessing well selected studies and providing data that adds to both IBS and IBD knowledge. I only have a couple comments/questions which should be considered minor:

  1. From Table 2, it appears that the great majority of subjects had inactive, quiescent or mild disease. Can the authors comment on whether the low disease activity created a ceiling effect on how much inflammation could improve on the diet.
  2. Are there any studies that assessed whether the diet is effective in decreasing functional GI symptoms in patients with active IBD (even mild) as compared to those in remission from their IBD?

Author Response

Dear Editor and Reviewers,

On behalf of my co-authors, we thank you very much for giving us an opportunity to revise our manuscript, we appreciate the editor and reviewers very much for their positive and constructive comments and suggestions on our manuscript entitled “A Low-FODMAP Diet Provides Benefits for Functional Gastrointestinal Symptoms But Not for Improving Stool Consistency and Mucosal Inflammation in IBD: A Systematic Review and Meta-Analysis”, with manuscript ID “nutrients-1711952”. The comments have been carefully taken into account and a newly revised submission has been uploaded. We highlighted all the altered passages in light yellow. The responses are as follows.

Response to the Reviewer 2,

Comment 1:

From Table 2, it appears that the great majority of subjects had inactive, quiescent or mild disease. Can the authors comment on whether the low disease activity created a ceiling effect on how much inflammation could improve on the diet.

Reply 1:                            

Thanks to the reviewers for their creative comments, this is an interesting point. We agree that low disease activity has a certain ceiling effect on diet improving inflammation levels. However, we would like to show that the original studies included in this paper are based on LFD improving functional gastrointestinal symptoms in IBD patients, so it is inevitable that most of included IBD patients are in low disease activity or remission, instead of severely active. In fact, LFD is not a treatment for patients with active IBD, and its use in patients with severely active IBD, not presented in any research previously, is even unethical. Meta-analysis can only be a secondary study on the basis of the original included literature, and we cannot subjectively select patients who are more suitable for the argumentation point of view. We greatly appreciate your valuable and divergent perspectives, which have helped to improve our manuscript. We set your comment as a thought-provoking question and add it to the revised manuscript, which may inspire readers conducting more comprehensive and extensive further research.

Changes in the text:

—See lines 330-332 in “Revised manuscript”.

Comment 2:

Are there any studies that assessed whether the diet is effective in decreasing functional GI symptoms in patients with active IBD (even mild) as compared to those in remission from their IBD?

Reply 2:

By reviewing the included articles and searching in the database, no such studies was found. It is possible that IBD population in mild disease activity or remission were both present in the same study, but no article compared the two population.

Changes in the text:

—None.

We have tried our best to revise our manuscript according to the comments. Attached please find the revised version, which we would like to submit for your kind consideration. 

We would like to express our great appreciation to you for your comments on our paper. Looking forward to hearing from you. Thank you and best regards.

Yours sincerely,

Jun Yi, MD, PhD & Xiaowei Liu, MD, PhD

Institute:

  1. Department of Gastroenterology, Xiangya Hospital of Central South University, Changsha, Hunan, 410008, China.
  2. Hunan International Scientific and Technological Cooperation Base of Artificial Intelligence Computer Aided Diagnosis and Treatment for Digestive Disease, Changsha, Hunan, 410008, China.
  3. National Clinical Research Center for Geriatric Disorders, Xiangya Hospital, Central South University, Changsha, Hunan, 410008, China.

Address:

Department of Gastroenterology, Xiangya Hospital of Central South University, Changsha, Hunan, 410008, China.

Tel:+86-731-89753268

Fax: +86-731-89753268

E-mail:junyee1989@csu.edu.cn & liuxw@csu.edu.cn
